# Social Problems in the Secondary Classroom: Gaps in Teacher Initial and Ongoing Training in the Andalusian Region of Spain from the Perspective of Intercultural Education and New Technologies

**Elisa Velasco, Mar Venegas and Kiko Sánchez-Miranda ***

Departamento de Sociología de la Educación, Facultad de Ciencias de la Educación, Universidad de Granada, 18071 Granada, Spain
* Correspondence: e.fjsasami@go.ugr.es

**Abstract:** The aim of this article is to analyze the initial and ongoing training of secondary school teachers to deal with the social problems of students in the classroom from the perspective of intercultural education and new technologies. The research starts from the thesis that these social problems can influence the academic and social success or failure of adolescents. Intercultural education and ICTs, with sustainability values, can favor the maturation process of students. The methodology used has been mixed: (1) documentary analysis of the curriculum of initial and continuing teacher training in relation to these social problems; (2) qualitative techniques with teachers, such as interviews and discussion groups to learn about their discourses regarding their professional vision and the training received. This has allowed us to understand professional practice from the perspective of intercultural education and new technologies. The results show training gaps in the curriculum for secondary school teachers to be able to work on social problems in the classroom with interculturality, together with the challenge of including new technologies and sustainability in the accompaniment of students. This research has done in Andalusia, the southernmost region of Spain, but with a desire to extrapolate its conclusions to other national or international contexts.

**Keywords:** social problems; adolescence; teacher training; intercultural education; sustainability; ICT; Andalusian

## 1. Introduction

In an increasingly diverse school, teachers need social and technological training that will enable them to meet the social problems of their students [1]. Working on individual differences from the perspective of intercultural, sustainable, and technological education seems to be an interesting way to face the great changes and challenges that can be found find in the classroom today.

The historical social problems of students in the adolescent stage are changing. The problematic situations of the classrooms of years ago are now different [2]. Specifically, the use of new technologies has increased considerably among adolescents, leading, on many occasions, to the improper use of ICT, which has led to new social problems that join the traditional ones in the educational system [3]. This use of technology can be an ally for teachers if they receive adequate training in intercultural education, appropriate use of ICTs and sustainability values applicable to all facets of their lives [4]. This training in educational innovation, together with social training for attention to social problems, has become indispensable for quality education in our secondary schools [2]. Thus, the aim of this article is to analyze the diversity of social problems present in the students of Obligatory Secondary Education (hereinafter OSE) and to support the importance of teacher training on this issue to provide them with the necessary tools to address this issue in the classroom from an intercultural education and a sustainable perspective.

The starting thesis is that the training received, both initial and ongoing, does not provide teachers with sufficient tools to address the social problems of secondary school students in the classroom [5]. In article incorporated the Intercultural Education and sustainability approach to our starting thesis. The study focuses on the autonomous community of Andalusia, located in the south of Spain. Andalusia is a region that has become increasingly diverse since the 1990s, with the arrival of a population of international origin.

### 1.1. Social Problems of Adolescent Secondary School Students

In this research, a social problem of secondary school students is understood as any social situation that negatively affects a significant volume of students, and that can impede the progress of these students in the personal, educational, and social spheres, making it more difficult than average for them to achieve school and social success [5].

It is understood that secondary school students have two conditions as a social group: as a school category, students; as a social category, adolescents. As a school category, a classic problem presents in the sociology of education and is derived from the material and cultural conditions of the student body and their families [6]. As a social category, secondary school students belong to the adolescent generation, between 12–18 years of age, so they face generational social problems specific to their age, where the fundamental variable is the strong feeling of belonging to the group of which they are part [7]. This consideration seems crucial to understand the way in which adolescent students manage their social problems. Hence, this research argues that it is essential to train teachers to provide them with key tools to work with social problems that can lead to school failure and, therefore, the social failure of students [2].

In the context of an increasingly diverse society, this problem requires a "situated" response from an intercultural and sustainable education open to diversity in the classroom [8]. School and social problems are intertwined. Additionally, the school is a privileged institution for the detection of risk factors that are difficult to identify at the family level. It is also a privileged place to learn other transversal competencies, which indirectly condition the educational quality and success, not only academic but also the social success of the students [9].

### 1.1.1. Social Problems of Secondary School Students at the Structural Level

When talking about structural problems affecting secondary school students, there are a series of objective variables and a legitimization of these problems that can cause students to fail academically and socially. The three issues that can affect this in a more direct way and that are considered to be closely related to the family environment are as follows:

- Poverty and Social Exclusion.

In the genesis of Poverty and Exclusion as a social problem [10], broader transformations of the social and economic structure intervene [6], which directly affect the academic and social success of the student body:

- Family educational level. The higher the probability that students are more likely to achieve greater success, the higher the educational capital of their parents [11].
- Family income. Having suffered economic hardship during youth increases the probability of experiencing poverty in adulthood [12].
- Family socialization. The distribution of roles in the family environment, motivation, and the valuation of effort for learning can be fundamental for the achievement of school success [6].

- Family model and structure: family violence, parental problems, single-parent families.

In this structural problem, the following variables must be taken into account: (1) practices related to the distribution of time and responsibilities and the distribution of domestic and family tasks. (2) Culture, concretized in attitudes and values when organizing family and work life. (3) The institutionalization and legislation regarding family law and family policies (the appearance of divorce and homosexual families, for example) [13]. At this

point, the fundamental premise for success in the maturation process of students is the closeness that parents have with their children, and the absence of conflict between them, especially in cases of divorce [14].

- Migratory Movements: Cultural or religious diversity is increasingly present in the classroom.

Globalization and population movements have led to knowledge and contact with diverse cultural and religious experiences that can lead to the search for dialogue and encounter [15] or to an increase in acts of intolerance or religious extremism [16]. In the classroom, attention to cultural and religious pluralism and diversity has not had a clear or continuous evolution [17]. Currently, the integration and accommodation of ethnic and religious minorities and their needs in school constitute a relevant phenomenon in Western societies [18], although there are still with quite a few gaps and points to move forward. Conflicts, acts of religious intolerance, and prejudice about other beliefs continue to occur in school contexts [15]. Theoretical and research advances in intercultural education explain that in order to achieve full inclusion, it is necessary to overcome approaches based on applying strategies to make certain people "fit into the ordinary" [19]. Working from an Intercultural Education, it is necessary to open the sustainable possibility that the centers, with their professionals, are the ones who try to adapt to diverse students, not only functional but also social.

1.1.2. Social Problems of Secondary School Students at the Generational Level

Toxic or violent friendships or couple relationships, difficulty in finding meaning in life with feelings of incomprehension and loneliness, emotional instability, or distrust in close people or in themselves can lead adolescence to school and social failure, in addition to emotional suffering that can favor isolation, depression, and even suicide [20]. It is important to accompany adolescent students by promoting the expression of communication and emotions and by training teachers to detect relationships that involve risky activities and preventing aggressive behaviors that are increasingly present in the classroom [20]. According to the literature review, the most common generational social problems are the following:

- Coexistence and school conflict, with bullying and cyberbullying as a high point.

A primary risk factor in the adolescent population is the relationship with the peer group, with bullying, especially cyberbullying, increasingly present in secondary classrooms, becoming present in 1 in 10 children, according to the organization Save The Children (2016) [21].

- Substance and non-substance addictions.

Traditionally, the cultural association between leisure and drug use as a means of acceptance by the peer group is one of the most worrying social and health problems in adolescence [20]. In addition, stressful and negative events, violent behavior [22], and problems in family relationships also pose an important risk factor for drug use among adolescents [23]. This consumption has a direct relationship with academic performance by disrupting cognitive functioning, diminishing academic responsibilities, damaging relationships with adults that influence academic performance, and exposing young people to unconventional behavioral norms [24].

Regarding the consumption of substances, there is the growing addiction of students to gambling and screens, which is currently one of the most prevalent addictions, according to data from Proyecto Hombre (2022) [3]. These new addictions to gambling and screens are leading many minors to absenteeism from school, conflictive relationships with their families and the educational center, and even committing crimes that allow them to continue accessing the consumption of addiction [3].

- Media and social networks.

The use of recent technologies appears as a cross-cutting variable in all of the social problems addressed. The increased use of social networks is a physical factor that affects the individual's emotional state, sense of humor, emotions, and school performance. The excessive and/or improper use of ICT can lead students to the different social problems discussed. However, they are a source of useful resources not only for students but also for teachers who can find an interesting tool to approach students and offer them existing alternatives through ICT [8].

- Eating disorders.

Eating disorders are behavioral alterations and have, as their objective, the effort to control an individual's weight and silhouette. Adolescents are the group most vulnerable to the development of eating disorders, pursuing an "ideal of beauty" imposed by Western culture [25]. Family abandonment and neglect are risk factors in both the personal and educational spheres, at a stage when they are in search of their identity, self-concept, and personal image and need to be cared for and listened to. If they do not find it in the family, they will look for it in the peer group and social networks. TICS are a vehicle for transmitting stereotypes that replace family roles, impacting the development of the person [26], providing a negative self-image of the body in the adolescent stage, and may even create an eating disorder.

- Specific Educational Support Needs (SEN).

In the latest educational policies of the Spanish State (LOE, 2006; LOMCE, 2013; LOMLOE, 2020) [27–29], the idea of inclusive practice in schools has been stressed. The LOMCE (2013) extends the term Special Educational Needs (SEN) to Specific Educational Support Needs. However, educational centers do not uniformly define the characteristics of SEN students. School learning communities favor the inclusion of students with functional diversity in terms of participation in shared tasks, acceptance within the group, and increased learning opportunities [30], but they do not include in their plans, programs, and projects the students who have other problems of a more social than educational nature, as it is included in the terminology of SEN students [31].

- Sexual and gender violence, absence of and rejection of affective-sexual education.

Since adolescence is a time of emerging sexuality and transition to adulthood [32], educational centers have an important role in the transmission of values that can guide students in this regard. From an emotional and affective point of view, sexual problems arise during adolescence due to the non-acceptance of diversity in sexual identity and orientation or to the rejection and the large number of risks associated with this area: gender violence, sexual aggression, unwanted pregnancies, forced marriages, harassment due to the non-acceptance of sexual diversity, or the violation of the sexual privacy of students on the Internet [20]. Some researchers have proven that homophobia and homophobic bullying are spread in schools to a large extent, something that is difficult to solve by teachers who lack the means and social training [9], sustainable values, and new technologies to intervene.

- Sustainability

Although the sustainability is not a generational social problem as such, but rather a generic problem that affects society, the faculty tries to convey the need for this approach [33] in a world of profound and rapid change [1]. It is a problem with which the adolescent generation is more proactive in intervening than other generations. Teachers should take advantage of this proactivity to work on it and thus be able to help the other social problems existing in the classroom [33].

## 2. Materials and Methods

As previously stated, the aim of this article is to analyze the diversity of social problems present in OSE students in order to support the importance of teacher training on this subject. In order to cover this research objective, a qualitative methodology has been designed as follows:

(1) To delimit the scope of the social problems of adolescent students in today's Andalusian society, the technique of documentary analysis has been used, temporally limited to the last decade and referring to the specific social problems of adolescent secondary school students. Documentary analysis is a technique based on the consultation of secondary sources whose purpose is to obtain data and information from documentary sources [33].

(2) In order to cover the review of the initial training of OSE teachers, the phase of the research from which this article originates was carried out in the time period between 2017 and 2021, undertaking the first analysis during the 2017/2018 academic year at the beginning of the research, and a second analysis in the last stage of the 2019/2021 academic year. In both cases, the data collection was based on documentary analysis, detailing each of the websites of the nine Master's degree programs of the nine public Andalusian universities that offer it and that are the subject of this study. The MAES curriculum has not been revised since it was implemented, so the data searched has not undergone significant changes since the web analysis was done. On the other hand, the COVID-19 pandemic did not bring any significant changes to the MAES curricula, beyond the fact that the contents were taught online.

These websites were considered to be the secondary sources that constitute the object of our analysis. The subjects of each module were studied from the perspective of the potential social problems for students that may arise in the classroom, making an exercise of critical analysis of the curricular programming of each subject by the university and searching exhaustively which curricular contents connect with the definition of social problems for students from which this research is based.

(3) The review of in-service teacher training was carried out in the 2020/2021 academic year. The data collection was based on the documentary analysis of the curricula that constitute the training offers of the 32 TTC that exist in the Andalusian region (Figure 1) through their websites in any of their modalities: courses, work groups, training in centers, tele-training, or other actions; always in relation to content focusing on attention to and working with students' social problems as defined in the theoretical framework.

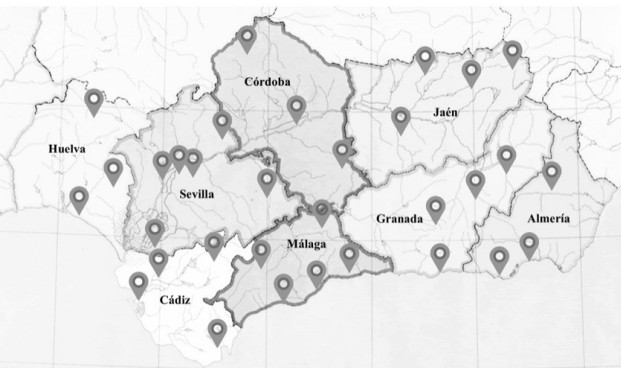

**Figure 1.** TTC of Andalusia by province.

(4) Finally, two qualitative data production techniques were used: the focus group and the interview with secondary school teachers. The focus group was carried out face-to-face with six teachers with professional experience in different educational contexts. On the one hand, teachers belonging to different types of schools according to: (1) their location in a rural or urban environment, (2) their ownership: public, subsidized, or private, and (3) the socio-economic level of the students. On the other hand, teachers had different characteristics according to: (2) their gender, (2) the type of subjects they teach, (3) the positions they hold in the school, and (4) the years of teaching experience. This technique pursues the production of a shared or collective group discourse through interaction, establishing conditions that enhance

the conversation among participants [34]. The differences between group members cease to be a problem or an inconvenience, becoming a resource or a mechanism for the production of group discourse, which intentionally aims to provoke and enhance them, stimulating interactions between them [34]. For this, it is necessary to prepare a guide outlining the topics to be discussed, the selection of the stimulus material for the animation of the group, and the convenient selection of people at the time of the composition of the group [35], seeking social heterogeneity.

For the interviews, teachers other than those in the focus group were contacted with the aim of expanding the sample but with the same selection criteria. The interviews allow us to access the individual and vital experiences of people, through which they can reveal their daily life and the social relationships they maintain [36]. It stimulates the flow of data and offers personal information that would otherwise be impossible to know. The ethnographic interview is characterized as a comfortable and well-directed conversation where the informant and the researcher relax, and everything takes the form of an approach and a natural encounter. In the ethnographic interview, the deep and implicit meaning of the interviewee's discourse is sought. The interview questions are indirect and open-ended, where the interviewer assumes a passive role to prevent the answers from being conditioned. In turn, the interviewer is in charge of controlling the direction of the interview, redirecting it in order to achieve the objectives set [37]. The list of questions asked in the interviews and in the focus group can be found in Appendix A. Table 1 shows the sociodemographic profile of all the teachers participating in the research, a total of 17.

**Table 1.** Subjects participating in the research.

| Acronym | Sex Assigned at Birth | Technique | Nature of Center | Specialty | Age | Years of Experience | Provinces | Territory |
|---|---|---|---|---|---|---|---|---|
| EMPUPV1 | Female | Interview | Public and Private Center | Economics | 33 | 8 | Granada, Cádiz, Málaga | Rural, Urban |
| EHPU2 | Male | Interview | Public Center | Geography, History and Ethics | 42 | 14 | Granada, Sevilla | Rural, Urban |
| EMPUPVC3 | Female | Interview | Public and Private Center | Biology, English | 28 | 5 | Granada, Sevilla, Cádiz | Urban |
| EMC4 | Female | Interview | Private Center | Technology | 40 | 12 | Granada | Urban |
| EMPU5 | Female | Interview | Public Center | Physical Education | 31 | 8 | Málaga, Granada | Rural, Urban |
| EMPU6 | Female | Interview | Public Center | Music | 55 | 28 | Málaga, Granada | Rural, Urban |
| EHC7 | Male | Interview | Private Center | Mathematics, Physics | 53 | 24 | Granada | Urban |
| EHPU8 | Male | Interview | Public Center | Mathematics | 41 | 15 | Almería | Rural, Urban |
| EMC9 | Female | Interview | Private Center | Language, French | 64 | 44 | Granada | Urban |
| EHPU10 | Male | Interview | Public Center | Language and Literature Latin | 66 | 43 | Granada | Urban |

**Table 1.** *Cont.*

| Acronym | Sex Assigned at Birth | Technique | Nature of Center | Specialty | Age | Years of Experience | Provinces | Territory |
|---|---|---|---|---|---|---|---|---|
| ECHPU11 | - | Interview with coordinator | Coordinator | - | - | - | - | - |
| GDMC1 | Female | Focus Group Discussion | Private Center | Computer Science | 36 | 11 | Huelva, Cádiz | Rural, Urban |
| GDHC2 | Male | Focus Group Discussion | Private Center | Mathematics | 31 | 6 | Córdoba | Urban |
| GDMPU3 | Female | Focus Group Discussion | Public Center | Biology | 40 | 14 | Almería, Granada | Rural, Urban |
| GDHPU4 | Male | Focus Group Discussion | Public Center | Music, Technology | 43 | 17 | Granada | Rural, Urban |
| GDHC5 | Male | Focus Group Discussion | Private Center | Language and Literature | 35 | 10 | Granada | Rural, Urban |
| GDHPU6 | Male | Focus Group Discussion | Public Center | Physics, Chemistry | 36 | 5 | Granada | Rural, Urban |

The material resulting from the interviews and the discussion group was analyzed through the qualitative analysis software Nudist Nvivo Release. In order to identify the questions that allow us to cover our research objectives among the teachers' discourses, the Grounded Theory procedure was followed. The emerging categories are identified in the following section.

Finally, during the last 7 years, a school autoethnography has been elaborated. Autoethnography is a qualitative research method characterized by linking the personal experience of the researcher with the social, political, and cultural concepts investigated [38]. It presents the originality of undertaking science from one's own experience, and at the same time, it implies a personal distancing, a "metapraxis" from a critical point of view that does not pretend statistical representativeness but reasonable examples of the educational context [38]. The choice of this technique is due to the fact that two of the members of the research team are themselves secondary school teachers.

**3. Results**

*3.1. Initial Training*

The Master's degree in teaching in Spain is composed of three modules: Generic, Specific, and Free Designation. The documentary analysis of the curricula of the three MAES modules in the nine universities analyzed in this study shows that the Generic module, composed of three subjects (Learning and Personality Development; Educational Processes and Contexts; and Society, Family and Education), is the one that in theory should address the social problems of students in greater depth. The reality is that the analysis of these subjects confirms that they do not cover the variety of problems presented in the introductory section or do so succinctly, and there are significant gaps in the training in social problems for secondary school teachers.

> *What I know practically is because I have been in contact since I was very young volunteering with teenagers. That is how I have learned to deal with these types of [social] problems with students. In my official MAES training, nothing at all.* (GDHC5)

The second module is the specific one. This also has three compulsory subjects to be taken for each of the Master's specialties (Complements for Disciplinary Training; Learning and Teaching of the corresponding subjects; and Teaching Innovation and Initiation to Educational Research) together with some other optional subjects. Most of the content of these subjects is focused on establishing pedagogical and methodological proposals for the subjects of the specialty in question, including specific didactics. The subject "Teaching Innovation and Initiation to Educational Research" has a part common to all specialties. However, it does not address any aspect related to the social problems of adolescent students, despite the fact that teaching innovation in the educational field is fundamental not only for the achievement of school success but also for the personal and social development of students. Innovation is limited here to the didactic–pedagogical but not to the social area of secondary education.

The other two compulsory subjects of this module have no relationship with the social problems of adolescent students, except for the approach to diversity. Diversity appears in a marginal way and is oriented to the scholastic success of students. However, it is not raised from the need to solve social, structural, and/or generational problems, as defined in this research. Some teachers insist on the idea that they do not need to have more knowledge about their specialty, but rather other types of content related to classroom intervention.

*I would omit subjects that are related to subject content, and I would put others that teach me how I have to act when faced with student problems in class.* (GDHPU6)

*I do not need them to give me Biology classes, or how I have to explain it. I need them to tell me what to do when I have a bulimic teenager in class.* (GDMPU3)

The rest of the elective subjects also do not provide training in relation to social problems. There is no content in relation to addictions, social networks and their danger, or eating disorders, for example. SENs are dealt with, although in a very specific way, in the educational field. Similarly, the brief reference to coexistence and gender diversity is focused on the smooth running of the class and not on the social problems of the students.

*I prefer to be guided more on how to tackle social problems and what they are. In my specialty, you already have the training, now you have to know how to apply it to human beings.* (GDHPU4)

Finally, the curricular review of the third module, the free configuration or designation module, shows some universities with subjects and content close to the subject that concerns us, but in others, there is no content at all.

*When there is a student with a problem, you do not know how to solve it. The first thing you do is to take the child and take him/her to the management, but you are not able to solve the problem yourself. I would have included all these things in the curriculum, even if it was in free configuration, to give you the tools to deal with these things in the classroom.* (EMC9)

The subjects close to the social problems that appear are those shown in Table 2 below. The few subjects in which social issues appear do not cover the need for training so that new teachers can work with the social problems of adolescent students. The content is synthetic, being a small part of the total subject. In addition, and as have pointed out, some social problems appear in some subjects of some universities; however, they are not worked on in all nine universities, but independently only in some of them.

**Table 2.** Free configuration subjects Initial Teacher Training by Andalusian universities.

| Free Configuration Subjects/Andalusian Universities | UCA | UGR | UMA | UAL | UCO | UHU | UJA | US | UPO |
|---|---|---|---|---|---|---|---|---|---|
| Attention to diversity and multiculturality. | | ✔ | | | | | | | |
| Education for equality. | | ✔ | | | | | | | |
| Towards a culture of peace. | | ✔ | | | | | | | |
| Management of school coexistence. | | | | | ✔ | | | | ✔ |
| Inclusive education and attention to diversity. | | | | | | | ✔ | | |
| Socio-cultural and intercultural aspects in the foreign languages classroom. | | | | | | | | ✔ | |
| Research and diagnostic methods. | | | | | | | | ✔ | |
| The challenge of diversity, disability, equality, gender, violence, and interculturality. | | | | | | | | | ✔ |

UCA—University of Cádiz; UGR—University of Granada; UMA—University of Málaga; UAL—University of Almería; UCO—University of Cordoba; UHU—University of Huelva; UJA—University of Jaén; US—University of Sevilla; UPO—Pablo Olavide University (Sevilla).

The implementation of the MAES was an important step forward in the training and professionalization of this group of teachers, bringing Spain closer to the EU and responding to the new ways of managing the education system from that need for greater teacher professionalization:

*Europe had demanded to do something because it was assumed that what was failing was the initial teacher training, and something had to be done... and it was done, and in that doing something, these masters arose, which I lived in first person, from my experience as a teacher of the same.* (EHPU10)

However, according to the results of our research, this improvement does not seem to be enough to qualify teachers to deal with social problems from an intercultural education perspective:

*I would have added a subject on the social reality in the classroom, what you can find, student profiles, problems in families that may exist, that give you the tools to be able to deal with it in some way and that you do not have to go totally blind when you encounter a class with such a culturally diverse student body.* (EMPU5)

*The skills that are developed in the Master's degree compared to the whole history of initial training designs that have existed in Spain in the last 40 years is obviously good. If we compare it with where we are going and above all... eh... how fast everything is moving and... how slowly things can change in an educational plan... frankly it can be improved.* (ECHPU11)

Thus, the analysis of the speeches of the participating teachers shows us the clear need they have to be trained in the social problems of the students. Practically all the teachers interviewed demand it, and most of them do so as a much-needed urgency. In general, they ask for less theory and less content related to their specialty in order to dedicate more time to social, pedagogical, practical, and sustainability issues in the day-to-day classroom.

*I would include topics to know how to deal with social problems: conflict mediation, equality issues, and... bullying, social networks, online games. Additionally, also topics that have to do with sustainability, because it really is also something new for which we are not prepared, to work on values in this sense, and that can help to address other problems that may arise.* (EHPU8)

*3.2. Continuing Teacher Training*

From the analysis of Continuing Teacher Training, the aim was to find out whether the offer provided by the public education system provides teachers with sufficient tools

to complete their capacity to respond to the social problems of adolescent students in the classroom on a daily basis from quality Intercultural Education and using sustainability values. Based on this analysis, it has inquired into the teachers' discourses on their own continuing education and their assessment of their ability to respond to the social problems facing students.

Of all the training actions offered by the Teacher Training Centers (hereinafter TTC), only those related to the social problems of the students or to related topics (e.g., emotional issues) were selected. Crossing these training topics proposed by the TTC and the number of the different training modalities carried out, it can be seen which topics are prioritized in the training of active teachers in Andalusia in terms of topics related to the social problems of students. Table 3 shows this.

**Table 3.** Social problems worked on in training actions related to students' social environment.

| Social Problem | Courses | Other Actions | Tele-Training | Working Groups | Training in Schools | Total | % |
|---|---|---|---|---|---|---|---|
| SNES | 84 | 16 | 2 | 53 | 12 | 167 | 15% |
| Gender, equality and education (co-education) | 52 | 19 | 1 | 34 | 2 | 108 | 9% |
| Sexuality | 11 | 11 | 0 | 5 | 3 | 30 | 3% |
| Cultural diversity | 13 | 6 | 1 | 9 | 4 | 33 | 3% |
| Religious diversity | 1 | 1 | 0 | 1 | 0 | 3 | 0% |
| Coexistence and inclusion | 32 | 35 | 1 | 35 | 31 | 134 | 12% |
| Bullying | 22 | 6 | 0 | 5 | 4 | 37 | 4% |
| Emotional field | 114 | 30 | 0 | 85 | 53 | 282 | 25% |
| Conflict mediation and management | 51 | 9 | 0 | 26 | 15 | 101 | 9% |
| Pedagogical innovation | 64 | 17 | 8 | 52 | 34 | 175 | 15% |
| Drugs | 2 | 0 | 0 | 0 | 0 | 2 | 0% |
| School early leaving and failure | 2 | 2 | 0 | 1 | 0 | 5 | 0% |
| Dysfunctional family | 3 | 8 | 0 | 0 | 1 | 12 | 1% |
| Social exclusion | 0 | 0 | 0 | 1 | 1 | 2 | 0% |
| First aid—health | 32 | 18 | 0 | 1 | 0 | 51 | 4% |
| Total | 481 | 178 | 13 | 308 | 160 | 1142 | 100% |

Prepared by the authors based on data from the TTC on their own websites. Training actions related to the emotional environment of secondary school students.

Emotional area and teaching innovation represent 25% and 15%, respectively, of the total number of training actions related to the area of social problems. These are the two areas to which most training space is currently dedicated. Although this is not direct training in social problems, it can help teachers in the smooth running of the classroom and students in the successful management of obstacles related to their maturation process. This is a wide margin of training actions that range from multiple intelligences to ICT courses, quality management, and professional or methodological updating, including the current proposals of learning communities, project-based learning, service learning, or cooperative learning. This is confirmed by the teachers interviewed, as will be seen below. It is striking that this area does not include the work on sustainability values, which is currently an area that is becoming increasingly necessary and aware in society and among our students.

*What predominates right now in TTC training are the topics related to bilingualism, new pedagogies, and other types of topics, let us call them transversal.* (EHPU10)

*Right now, there is a lot about Moodle platforms, digital teaching . . .* (EMPUPVC3)

*I think that in general, they have prioritized a lot the New Technologies, the topic of ICTs, Moodle platforms, Classroom, and even more now. More so now, after teleworking and virtual classes and so on. I think they have focused, above all, on that line.* (EMPU5)

SEN, as well as Educational Innovation, have 15% of the total number of training actions related to the social environment of students. This is training to work with students

in need of special and specific educational support. That is, students with high abilities, immigrant students with language support needs, or with learning difficulties due to physical or cognitive functional diversity, or with social problems, which also includes students with ASD or ADHD. The orientation that is usually given to the training actions that are framed in this section is almost exclusively for students with learning difficulties in the physical, cognitive, or sensory aspects without including the social part of SEN recognized in the law.

> *For example, diversity, ADHD, children who have difficulties, well, we have focused on those children, we do have training. However, it is true that we have focused on working techniques in the classroom on this type of diversity and not on social diversity.* (EMC4)

The highest percentage of PA directly related to social problems is centered in the section on school coexistence and conflict (21%). Within this section, it could be divided into two themes: attention and work for good coexistence (12%) and conflict resolution (9%). Related to this topic is bullying, a much more specific and specialized topic in terms of conflict (4%). It has chosen to present the percentages of this section separately because it has been found that many of the proposed PAs are aimed at the creation and management of coexistence classrooms. Preventive or interventionist work in schools has been left aside. Despite being such a topical and socially important issue, it is curious how common it is in the interviews to hear comments on the lack of training in this area and that, on many occasions, it is left to personal concerns:

> *As for conflicts and so on, well, it is true that we go a bit on a daily basis, so of course, we are working on a daily basis and working personally, so, I do not know, it could be more training.* (EMC4)

Likewise, in the research, there were teachers who have referred to the desire to receive training related to this topic, and more specifically with bullying, understood as a much more aggressive and negative situation.

> *I would like to be trained, it has always caught my attention, on the subject of bullying, cyberbullying, it has always caught my attention, I tell you, because it is something that I do not tolerate in my classes. I would like to see more offered on the social problems of bullying, cyberbullying, and problems of dysfunctional families.* (EMPU5)

The training focused on coeducation is 9%. The field of health, 5% present with some isolated mention of the topic of eating disorders. More residual, the topics centered on cultural or sexual diversity appear, with only 3% each. It should be remembered that the percentages refer to the social field and not to the total number of training actions offered by the TTC. Family matters account for 1%. School failure, religious diversity, drug addiction, and social exclusion represent 0% of the total number of actions carried out for the training of secondary school teachers in social problems. The reference to these gaps in the ongoing training of teachers is also found in their comments:

> *We have been dealing a lot with diversity in the classroom, especially ADHD, but in a specific way, that is, we are given the guidelines to work on how to lead in that aspect, but regarding everything else, in terms of emotional issues, addictions, conflicts, disorders, and gender equality, we do not go much into coeducation in that line.* (EMC4)

> *To solve social problems and so on, I think that the continuing education I have done has not given me tools because they were not to do with that topic.* (EMPU6)

Despite this, the teachers interviewed are not usually proactive in requesting training in social problems, even though they see the need for tools in this regard on a daily basis. One of the main reasons given by teachers for not requesting this type of training is the amount of bureaucracy in the educational system and the workload of teachers, especially in private schools. The elaboration of documentation, reports, minutes, and bureaucracy, in general, involves the dedication of a good part of the teacher's time and energy, leaving very limited possibilities to dedicate to the training itself.

*The TTC makes some offers, but I eliminate them, because I do not have time between all the tasks and the bureaucracy I have to carry out.* (EMPUPVC3)

*It is true that I recognize that, although it is a tremendous hassle to combine it, because then you have to give classes, write papers and have tutorials, and it does not fit in.* (GDHC5)

The offer received from the TTC is often considered "meaningless", not adjusting to the real needs of daily work in the school and in the classroom. There is a feeling that training actions are offered on topics not related in any way to the educational field. Some also refer to a lack of "marketing" to motivate teachers to carry out training actions.

*If you go to the TTC website, you will see the courses that are available. It is true that better marketing could help me to say, look, I have to do it. Besides, sometimes the offer is not adjusted to our needs, being focused on things that have nothing to do with education.* (EMPUPVC3)

*However, the rest are, I do not know, they are all, are not they? "Astrophysics in the classroom", such, but for me the same, "camera lights and action", ehh aaa I do not know, there was one that I was crazy about, it was about massages or something like that; this, comprehensive therapy to the lymphatic system and elements of the thick ulcers. I am not kidding, I am reading it. These are topics that in the end do not solve your work in the classroom at all. They talk about super innovative and super transgressive things, but then they do not go to the problem.* (GDHPU6)

*The most common thing is that the center, through the teacher in charge, informs the teaching team of the training offered by the TTC of reference. However, normally there are not many, nor do we propose training proposals adjusted to training in social problems or in sustainability, for example.* (EHPU2)

*Well, I would have liked some training in, in that, right? In social problems. Additionally, then that, in ongoing training we are not given any; in the end I think that training is given to you by many colleagues.* (GDHPU6)

## 4. Discussion

According to the largest survey of young people in Spain (11,854 young people), (citing the future is now) problems of a social nature occupy fourth place in the ranking of the top 20 concerns of young Spaniards. The complexity and speed of the changes affecting the diverse secondary school student body pose significant challenges [1] in initial and continuing teacher training.

The secondary school teachers of our time have to face a student body marked by social problems that, especially at present, have a great specific weight in their school and social performance [10]. This wide range of social, structural, and generational problems provides us with evidence that refers us to the focus of this article: the great diversity that characterizes secondary students in today's schools. Hence, intercultural education is, together with sustainability and new technologies, two transversal strategies of interest to address the social problems of students in the classroom.

If it does not take care of both the initial and ongoing training of teachers in terms of achieving the necessary tools to work on the social problems of students; it will be more difficult for them to carry out their teaching tasks with an educational quality that is committed to interculturality and sustainability as an educational style. This may have consequences in the failure of the students and in the professional's own work performance. In this sense, collaboration with families is one of the most important components of teacher satisfaction. However, also in this area, the training offered in relation to this topic is practically null [2].

Teacher training can facilitate reflection on one's own teaching practice in order to respond to problems in the daily life of the classroom [5], incorporating the transversal vision of sustainability [4]. This will allow for more and better interdisciplinary work in

secondary schools, where the qualification of teachers to work on the social problems of their adolescent students may be the first step towards the necessary interprofessional attention to these problems, especially in the context of the current decomposition and recomposition of teaching work [7].

The responsibility for the lack of teacher training in social problems does not have a single or major cause. The legislation in force, the university, the teaching staff, the motivation of the trainees, and the centers that receive trainee teachers are all part of the problem but also part of the solution. The creation of spaces of convergence between the different agents involved in the matter could open paths of understanding and improve the quality of future teachers and, thus, the students as members of our society [5]. These spaces could be the seed of changes in the needs demanded by a good number of members of the teaching staff with respect to the real option of times and spaces to facilitate quality continuing education.

## 5. Conclusions

Empirical data from fieldwork show that the training of secondary school teachers in the Andalusian autonomous community is still insufficient for appropriate attention to social problems [9]. Including this topic specifically in the teacher initial and ongoing training would contribute to the improvement of teacher professionalization, with a greater capacity to respond to the social problems of adolescent secondary school students [2,5].

It is important that teachers have space and time for training in the social sphere to enable them to be able to work from classroom observation, active listening, conflict resolution, and more personal work with students instead of being focused primarily on the bureaucratic aspects of the teaching task [39]. There is, therefore, convenience and a need to increase the quality and quantity of training for new and senior teachers in quality intercultural education based on the cross-cutting values of sustainability.

The implementation of the MAES in Spain is part of the Spanish Government's intention to adhere to the European Higher Education Area (Bologna plan), being accredited at European level. Therefore, the data provided in this study could be extrapolated to other European contexts. Even so, it would be interesting to compare the data from this study with initial and, above all, ongoing training in other European countries in order to draw more conclusions in this respect. This could be the focus of further research. On the other hand, another limitation of our work would be to increase the number and characteristics of the people interviewed.

**Author Contributions:** The author's contributions are: Conceptualization, E.V. and M.V.; methodology, E.V.; software, K.S.-M.; validation, E.V., M.V. and K.S.-M.; formal analysis, E.V.; investigation, E.V.; resources, E.V.; data curation, E.V.; writing—original draft preparation, E.V.; writing—review and editing, M.V.; visualization, K.S.-M.; supervision, K.S.-M.; project administration, K.S.-M.; funding acquisition, E.V. All authors have read and agreed to the published version of the manuscript.

**Funding:** This research received no external funding.

**Institutional Review Board Statement:** Not applicable.

**Informed Consent Statement:** Informed consent was obtained from all subjects involved in the study.

**Data Availability Statement:** Not applicable.

**Acknowledgments:** Materials used for the Proyect I + D + i ROMANCE SUCC-ED (Ref.: B-SEJ-332-UGR20) and TraSPASA (Ref.: PY20_00198).

**Conflicts of Interest:** The authors declare no conflict of interest.

**Appendix A. Summary of the Interview's Script for Secondary School Teachers in the Andalussian**

1.  Profile of the teacher, and description of the center's environment (physical, social, cultural, religious, economic context, presence and profile of residents of foreign origin and gypsies, etc.).

    -   Has there been, and is there currently, a significant presence of students with social problems in the classroom?

2.  Social problems of secondary school pupils, and responses offered by the educational center.

    -   What do you consider to be the current social problems faced by secondary school pupils in the classroom?
    -   Do you think that these problems affect girls and boys equally?
    -   Have you noticed any change in the social problems presented by pupils since you have been teaching?
    -   How have you experienced this change, and how do you think your fellow teachers have experienced it?
    -   What attitudes do teachers at your school have towards these issues? Examples.
    -   Is there a significant difference in the social problems presented by students depending on whether the school is rural or urban?
    -   Do you feel that schools are responsive and/or able to respond to these social problems? What measures have been implemented during your professional experience and in your current school?
    -   Do all pupils in the school have the same opportunities for success at school, regardless of whether or not they have any kind of social problem? to delve into factors specific to the family; factors specific to the pupil; factors specific to the structure and the educational system; factors specific to the school and its professionals.

3.  Initial teacher training. Initial training received for teaching practice with regard to the attention and work with the social problems presented by pupils.

    -   Do you think that the initial teacher training received through the MAES is a training that prepares the teacher sufficiently for day-to-day professional practice in general? Do you consider the training to be adequate in terms of content and time?
    -   And for professional practice, do you consider the internships you receive to be sufficient in terms of content and duration?
    -   Do you think that the initial training you received to become a teacher prepared you to face the social problems you have encountered in the classroom? Why?
    -   In relation to which social problems do you consider that you have training, and in relation to which you do not?
    -   After your experience as a teacher, what aspects related to students' social problems would you have added to the initial training curriculum? Why?
    -   Do you think it is important and necessary for teachers to be prepared during initial teacher education to deal with pupils' social problems? Why?

4.  Ongoing training for secondary school teachers in the Andalussian.

    -   Do you know the Teacher Training Centers (TTCs) as centers for access to in-service teacher training? Does your educational center work with its corresponding TTC?
    -   What aspects of ongoing training have you focused on during your professional career?
    -   What options have been prioritized in your choice of ongoing training by the administration, or by the school to which you belong? And from your own training concerns?

- Do you think that the courses you have received have provided you with the necessary training to tackle the social problems of pupils? In what aspects would you like to be better prepared?
- What actions are carried out in your school to promote the training of the teaching staff (talks, courses, etc.), both external and internal?
- The different Andalusian TTCs offer online training in the following areas of work with social problems, some of which are closely related: Coeducation (gender, equality, and education); sexuality; sexual diversity; cultural diversity; religious diversity; coexistence and exclusion; school bullying; emotional environment; conflicts and conflict mediation; pedagogical innovation; family/destructuring and exclusion; addictions; school failure and dropout; avalanche (problems with food). Did you know about this offer?

5. Education policy and accountability.

- Could you assess current education policies and their impact on teachers' work, and on the quality of their work?
- What do you think about the recognition and social image currently projected on teachers in society?
- Do you know the current education law? Do you consider that any account is taken of the initial and in-service training needs of teachers in relation to, or to face the social problems of pupils?

6. Reality of professional teaching practice.

- In your years of teaching, do you think that the management teams have made sufficient effort to try to deal with the social problems of pupils through the attention that teachers can offer their pupils, for example, through tutorial action?
- Who do you think has the task of attending to, responding to, and managing the social problems of pupils at the school?
- Is it the role of the teaching staff to attend to or manage the social problems of pupils?
- What do you think are the needs of the teaching staff, and what can the educational structure cover, in order to attend to and work with the social problems of pupils?
- Does the general dynamics of your day-to-day teaching allow you to dedicate time and energy to understanding and dealing with the social problems of your pupils, and to responding to these problems?
- What would you change in the organization of your school and your teaching practice in order to respond more effectively to students' social problems?
- From your academic experience, do you consider the relationship between students' academic results and the social problems they may be experiencing to be important? In what way do you think they may be related?
- Are you satisfied with your teaching task in terms of the attention you give to the social problems of your students? What do you think you aren't doing correctly and where do you think you are getting it right? What could you improve in your professional teaching practice?

7. Other professionals in the school.

- Are there any intra-school professionals who come to the school to work or provide guidance on how to deal with the students' social problems? How do they rate them?
- What professionals from outside the center collaborate with the center? What functions do they do?

8. Families.

- Involvement and forms of participation of families in relation to the social problems of their children, which, on many occasions, are closely related to the family situation.
- Assessment of their different forms of involvement/participation and proposals for improvement.
- What are the specific channels of communication with pupils' families when there is an obvious problem?
- Do you carry out, from your center, any activity aimed at working with the family in relation to the social problems of the pupils? What valuation do you make of these actions?
- Explain in more detail any significant experience which you value most positively.
- What is your perception as a teacher of the role and the responsibility assumed by families in relation to pupils' social problems?

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
