# Peer review of "Social Problems in the Secondary Classroom: Gaps in Teacher Initial and Ongoing Training in the Andalusian Region of Spain from the Perspective of Intercultural Education and New Technologies"

_sustainability, doi:10.3390/su15010339_

Round 1

Reviewer 1 Report

Dear Authors, congratulations on your relevant and pertinent work and bringing forth such stringent issues in secondary education and not only. 

The value of the paper lies in identifying and uttering the need for consistent teacher training in tackling present day issues that the youth of today is facing towards their academic/ professional success and well-being.

The contextualization of the subject is pertinent, the terms and notions are disambiguated clearly in the first parts. 

It is nevertheless of the essence that the authors consider revisiting a few issues in the manuscript towards its potential future publication:

1. The title: it seems that it lacks comprehensiveness and it sounds a bit ambiguous, in the sense that without reading the paper, one would think that diversity stems from intercultural education and new technologies (potentially include 'from the perspective of....')

2. The formality and quality of the writing should be considered (for instance the significant use of the first personal plural pronoun we that imparts a shade of subjectivity to the matter at hand, especially when qualitative research is undertaken, 'seems to be an interesting way' (line 28), problems that join traditional ones (line 35)). More academic and impersonal tools of expression and language are kindly recommended to professionalize the writing a bit further;

3. There is an extensive and imbalanced approach of the social problems at structural level. The part that they cover takes a significant share of the paper. They, themselves, are granted more or less space in the economy of the paper (according to importance? occurrence?). What is the rationale behind adding sustainability there?

Could the authors consider a different organisation of this part? Is there any relation with teacher curriculum content? to their experiences as depicted from the interviews? 

4. lines 231-232, the authors could consider a more detailed approach of the different contexts in between the brackets, as shown in the Table 1. 

5. Table 2 should be in English for consistency and coherence. 

6. There is no Conclusions and Recommendations part (perhaps the last 2 paragraphs) and most importantly, there is no Limitations part that I would suggest the authors to include. 

Good luck with your work

Reviewer 2 Report

The purpose of this article (“The aim of this article is to analyze the initial and ongoing training of secondary school teachers to deal with the social problems of students in the classroom from the perspective of intercultural education and new technologies) sounds very timely and practically significant for contemporary secondary education.

The authors presented a very good analysis of the secondary school students’ social problems in the first part of the article. But the presented analysis, firstly, covers a wider range of social problems than those stated in the title (i.e., not only the problems of intercultural education and the use of ICT), and, secondly, in the second part of the article, these problems are considered on the example of only one region of Spain (the Andalusian region).

In this regard, I would like to invite the authors to specify and change the title of the article to Social problems in the secondary classroom: gaps in teacher initial and ongoing training in the Andalusian region of Spain”.

If the authors accept this proposal, then it will be necessary to adjust the purpose of the article and its description in the abstract. In any case, I believe that in the title, in the abstract, in the keywords and in the text of the first part of the article, it should be mentioned that the analysis of the problem was carried out on the example of the Andalusian region of Spain.

I also have a few smaller comments and suggestions for improving the article:

1   1) The authors write in the abstract that a mixed methodology was used. It is common to call mixed methodology the use of quantitative and qualitative methods. However, the study is dominated by qualitative methods, although content analysis could be applied to analyze the content of curricula and the content of websites

         2) The authors indicate that they analyzed the problem from 2017 to 2021 and the final stage of the analysis included the 2020/2021 academic year, but they do not mention that lockdowns were introduced worldwide in this academic year due to the COVID-19 pandemic. it can be assumed that the situation of the pandemic affected, for example, the content of the websites of the Universities and TTC

     3) Table 2 needs to be translated into English

     4) In the Discussion section (or in the Conclusions section?), I would like to see more specific findings of this study, its limitations, perspectives, as well as its practical applications.

Reviewer 3 Report

The peer-reviewed manuscript addresses a very important and topical issue of quality education.

In my opinion, however, significant improvement and supplementation of some important information is required before the publication of the manuscript.

Detailed notes:

1. Are pedagogical study programs at universities subject to a systemic improvement process by identifying the needs of external stakeholders (high school representatives, high school students, parents).

2. Are study programs at universities internationally accredited? Is it up to date?

3. How often are curricula content and changes in pedagogical studies curricula verified?

4. Is it possible to compare Spanish solutions with solutions in other European countries?

5. The materials and methods section should be supplemented with more detailed information about the questions asked. (research methodology is to enable the replication of the conducted research).

6. Is the research sample of 17 people represented? In my opinion, the size of the research sample is a certain limitation. The authors should define the limitations resulting from the adopted research method as well as the sample size.

7. In the section, the results should be translated into English table 2.

For what period do the data from the TTC websites come from?

In my opinion, the presented results are not legible and clearly defined.

8. I propose to separate the Discussion sections into Discussion and Conclusions.

The article has potential, I hope the authors will improve the manuscript.

Round 2

Reviewer 1 Report

Congratulations to the authors for improving the manuscript and considering reviewers' comments and recommendations.

Reviewer 3 Report

In the new version of the manuscript, the authors took into account my comments as well as those of other reviewers. The methodological part is clear and understandable. I recommend the article for publication.